# Generation of a Normal Long-Term-Cultured Chinese Hook Snout Carp Spermatogonial Stem Cell Line Capable of Sperm Production In Vitro

**DOI:** 10.3390/biology11071069

**Published:** 2022-07-18

**Authors:** Xiao Chen, Yuting Kan, Ying Zhong, Muhammad Jawad, Wenbo Wei, Kaiyan Gu, Lang Gui, Mingyou Li

**Affiliations:** 1Key Laboratory of Integrated Rice-Fish Farming, Ministry of Agriculture and Rural Affairs, Shanghai Ocean University, Shanghai 201306, China; m190100049@st.shou.edu.cn (X.C.); m200100064@st.shou.edu.cn (Y.K.); zhongy10@haid.com.cn (Y.Z.); jawadktk1293@gmail.com (M.J.); m200100082@st.shou.edu.cn (W.W.); m210100083@st.shou.edu.cn (K.G.); 2Key Laboratory of Exploration and Utilization of Aquatic Genetic Resources, Ministry of Education, Shanghai Ocean University, Shanghai 201306, China; 3Key Laboratory of Microecological Resources and Utilization in Breeding Industry, Ministry of Agriculture and Rural Affairs, Guangzhou 511400, China

**Keywords:** *Opsariichthys bidens*, spermatogonia stem cell line, in vitro spermatogenesis, differentiation, cryopreservation, cell transplantation technology

## Abstract

**Simple Summary:**

In vitro-induced differentiation of sperm cells is a key technology for genetic resource conservation. In the past ten years, *Opsariichthys bidens* has become a famous and excellent aquatic species in some areas in China. However, its genetic resources have reduced dramatically. To protect against the decline of *O. bidens*, a long-term-cultured spermatogonial stem cell line (ObSSC) of adult *O. bidens* was successfully established. The result of our study showed that ObSSC had a diploid karyotype and stable growth over more than 2 years, with SSC-typical gene expression patterns. Furthermore, our research demonstrates the potential and regulation mechanism of fish spermatogonial stem cell differentiation into different cells of three germ layers. Our findings will assist further research on the genetic resource conservation of germplasm in a commercially and ecologically valuable fish species.

**Abstract:**

*Opsariichthys bidens* belongs to the family *Cyprinidae* and is a small freshwater economic fish widely distributed in China. In recent years, the natural resources of *O. bidens* have been drastically reduced due to overfishing and the destruction of the water environment. The in vitro culture and long-term preservation of germ stem cells are the key technologies to keep genetic resources from degeneration. However, except for the establishment of the first long-term cultured medaka spermatogonia cell line (SSC) capable of producing sperm in vitro in 2004, no other long-term cultured SSC line has been found in other fish species. In this study, we successfully established another long-term-cultured spermatogonial stem cell line from *Opsariichthys bidens* (ObSSC). After more than 2 years of culture, ObSSC had a diploid karyotype and stable growth, with the typical gene expression patterns of SSC. Under in vitro culture, ObSSC could be induced to differentiate into sperm and other different types of somatic cells. In vivo, ObSSC could differentiate into different cells of three germ layers upon being transplanted into zebrafish embryos. Our research helps to explore the potential and regulation mechanism of fish SSC differentiation and spermatogenesis in vitro, provides a new way for solving the problem of fish genetic resource degradation and lays a foundation for further research on fish germ cell transplantation.

## 1. Introduction

Fish have the utmost diversity of all vertebrates, with over 32,000 species recorded, which is half the number of vertebrates on earth. Several fish species are facing extinction due to rapid population declines [1]. As a result, the long-term preservation of fish genetic resources is becoming extremely important for the protection of fish species with declining populations [2]. Due to the limitation of available methods for cryopreservation of fish eggs and embryos [3], cryopreservation of germline stem cells (GSCs) and its successive transplantation into donor fish is an alternative option to retain the genetic resources of endangered fish [4,5].

The establishment of germline stem cell (GSC) cultures, such as primordial germ cells (PGCs), spermatogonia stem cells (SSCs) in the testis, and oogonia stem cells (OSCs) in the ovary, have attracted considerable interest [6,7]. SSCs are abundant in immature testis during all phases of testis development, and a method for efficiently isolating SSCs in many species has been developed. In contrast, PGCs and OSCs are difficult to isolate [8].

Currently, many culture systems have been designed to imitate the germ cell niche and give various supplementary factors for SSCs to make haploid sperm [7]. In mice, the establishment of in vitro-derived gametes was performed by stepwise differentiation from pluripotent stem cells after exposure to morphogens and hormones in coculture systems [9,10,11]. In zebrafish, early-stage spermatogonia and Sertoli-like cells were cocultured in adherent culture systems to produce functional sperm [12]. In the cyprinid honmoroko (*Gnathopogon caerulescens*), from dissociated testicular cells in suspension culture, functional sperm were obtained [13]. In swamp eel (*Monopterus Albus*), enriched SSCs were isolated and cultured in vitro from the testis [14]. In marine four-eyed sleepers (*Bostrychus sinensis*), premeiotic spermatogonia were obtained and further flagellated sperm from spermatogonia in a 3D culture system [15]. Our laboratory previously successfully established a normal SSC line (SG3) from the adult testis of medaka (*Oryzias latipes*), which can be cultured for a long time in vitro, and successfully made the SG3 cell line recapitulate the spermatogenesis process and produce motile sperm in vitro [16]. However, due to the lack of an efficient culture technique that supports continuous SSC proliferation in vitro, stable SSC lines in other fish species have not been recorded [17]. Therefore, establishing a culture system for long-term in vitro SSC lines in aquaculture fish remains a key challenge.

*O. bidens* is becoming an increasingly important small aquaculture species in China, with high economic value and wide distribution. *O. bidens* wild stocks have declined due to environmental changes and overfishing. Its reproductive biology research should be carried out to protect the diversity of *O. bidens* genetic resources. Recently, we performed an analytical study on the *O. bidens* gonad transcriptome [18]. In the present study, we successfully established a long-term-cultured spermatogonia cell line capable of sperm production from the adult testis of *O. bidens* in vitro. *O. bidens* SSCs can be induced to differentiate into a variety of somatic cells in vitro and in vivo. This study helps to provide a new model for the study of fish SSCs and lays the foundation for further studies on the reproduction of *O. bidens.* It also provides a donor for further research on germ cell transplantation.

## 2. Materials and Methods

### 2.1. Fish

Fry fish were obtained at Jinhua Hengyuan Agricultural Science and Technology Co., Ltd. (Jinhua, China) and were transported to Shanghai. They were raised (26 °C) in an automated breeder with a 14 h:10 h light/dark cycle at the Shanghai Ocean University *Opsariichthys bidens* Center (Shanghai, China). The study was performed following the Declaration of Helsinki and approved by the Shanghai Ocean University Animal Care and Use Committee with Approval Number SHOU-2021-118.

### 2.2. Preparation of Culture Medium

Preparation of culture media ESM2 and EMS4 was carried out as described [19]. The basic fibroblast growth factor (bFGF; Gibco, Carlsbad, CA, USA) and medaka embryo extract (MEE) content of ESM2 is higher than ESM4. For ESM4, each 1 L of DMEM/HEPES (pH 7.8; Gibco, New York, NY, USA) contains the following components: FBS (150 mL; Gibco, NSW, Australia), MEE (1 mL; 400 medaka embryos/mL; 2.5 mL for ESM2), fish serum (10 mL; Seabass; serum extraction method as described [16]), bFGF (20 µL; 0.1 mg/mL; 100 µL for ESM2), L-glutamine, nonessential amino acids, sodium pyruvate, and pen/strep (all 100×; 10 mL; Gibco, New York, NY, USA), 2-mercaptoethanol (4 mL; stock 50 mM; Sigma, Burlington, MA, USA) and sodium selenite (1 mL; stock 2 µM; Sigma). The culture medium was filtered using a filter (0.22 µm) and stored (4 °C) for up to 6 months.

### 2.3. Primary Culture and Subculture

All equipment used in the experiment was sterilized before use. The 6-month-old fish were placed on ice for 15 min and sterilized with 75% ethanol. The testis was dissected carefully and washed 3 times in phosphate-buffered solution (PBS) containing 1% penicillin/streptomycin. After being finely minced, the testicular tissues were subjected to 1 h dissociation (28 °C) in a solution including 5% trypsin (25 μL; Worthington, OH, USA; dissolved in L-15 medium), 40 mg/mL collagenase H (50 μL; Roche, Mannheim, Germany; dissolved in L-15 medium), FBS (25 μL) and 1% DNase I (25 μL; Roche, Mannheim, Germany; dissolved in L-15 sterile water) in 500 μL L-15 for 1 h at 28 °C. Next, the dissociated cells were transferred to gelatin-coated 6-well plates and cultured (28 °C) in a culture medium (1.5 mL). Cells were cultured at 28 °C in air. Cell subcultures were performed as described [16]. Briefly, dissociated cells were cultured in EMS2 medium for several weeks, passaged to Passage 15, and stably proliferating ObSSCs were obtained. From Passage 15 of the culture, ObSSCs were cultured in fresh ESM4 medium, and the cells were passaged to 50 passages and named the ObSSC line, which has more than 100 passages in 2 years of culture.

### 2.4. Chromosome Analysis

ObSSCs (at Passages 20 and 60) were used for chromosomal analysis. The ObSSCs were seeded into a 6 cm culture dish for culturing (24 h, 28 °C). Cells with active growth were cultured (4 h, 28 °C) in colchicine solution (20 μg/mL) to accumulate metaphases. After 2 PBS washes, cells were treated with 0.25% trypsin/EDTA solution (Gibco, NY, USA) for 30 s for dissociation, which was then removed, followed by the addition of PBS (1 mL). Cells were put into microcentrifuge tubes (1.5 mL) and gradually added with hypotonic solution (1 mL, 2.98 mg/mL KCl) for 30 min incubation for cell swelling. After that, the hypotonic treatment was terminated by adding fresh fixative (0.2 mL, 3:1 methanol/acetic acid). Next, following resuspension in 1 mL fixative, the cells underwent 30–60 min fixing after two fixative changes. After cell centrifugation and resuspension in fixative (0.1–0.5 mL), cells were loaded onto clean glass slides; 5% Giemsa solution was used for chromosome staining (15 min, room temperature). Slides were photographed and examined through a Nikon Eclipse 80I fluorescence microscope.

### 2.5. Cryopreservation, Recovery, and Alkaline Phosphatase (AP) Staining of ObSSC

The 4 × 10^5^ ObSSCs were resuspended in 1.5 mL medium containing 15% FBS and 10% dimethyl sulfoxide (DMSO). Cells were transferred into 2 mL cryogenic vials (Corning, Reynosa, Mexic) and placed in a cell freezing container (Corning, Reynosa, Mexic), where they were incubated at (−80 °C, 48 h) before storage in liquid nitrogen. To recover the cells, they were immediately thawed (37 °C, 2 min) in a water bath being centrifugation (1000 g, 10 min). Next, cells were seeded into a gelatin-coated 24-well plate cell after being suspended in fresh subculture medium. After 24 h of incubation (28 °C), the medium was replaced. Part of the cells was taken and placed into a gelatin-coated 24-well plate, and AP staining was prepared after the cells adhered to the wall. AP staining was performed as described [20].

### 2.6. Extraction of Total RNA and RT-PCR

To isolate RNA, cells were collected from 4 wells of a 24-well plate and treated with Trizol (Sigma, Burlington, MA, USA). Total RNA was precipitated in isopropanol after chloroform extraction, followed by 75% ethanol washing and then dissolution in 10 μL nuclease-free water for cDNA synthesis. cDNA was synthesized from 1 µg total RNA (TaKaRa, Kusatsu, Japan) using the M-MLV Reverse Transcription Kit (TaKaRa, Kusatsu, Japan). To obtain cDNA fragments, *O. bidens* gonad raw sequence data were used [18]. Primer sequences are shown in Table 1. PCR amplification conditions were: 95 °C for 5 min, and then 35 cycles of 95 °C for 20 s, 58 °C for 20 s, and 72 °C for 30–60 s in a 20 μL system, followed by 72 °C for 10 min. β-Actin was used as an internal reference.

### 2.7. Immunofluorescence Staining of ObSSCs

The round 24-well cell climbing slices were soaked overnight in 75% alcohol before being covered with gelatin. ObSSCs were seeded on 24-well climbing slices, and the media were removed after the cells reached 70% confluence. After 2 PBS washes, cells were fixed (10 min) in 4% paraformaldehyde. Following PBS washing, the cell climbing slides were transferred to a new 24-well plate with the cell-covered side facing up. Cells were subjected to 10 min 0.1% Triton-X 100 (200 μL) treatment and then 2 PBS washes. Next, cells were blocked (30 min) with 3% bovine serum albumin (BSA, 200 μL, Sigma) and washed with 500 μL PBS once. Next, the cells in one well were incubated with 200 μL Vasa antibody (1:200 dilution with 3% BSA; Invitrogen, Waltham, MA, USA, PA5-30749; this antibody corresponds to a recombinant fragment of human DDX4; Gene ID: 54514), while the cells in another well were added with 200 μL proliferating cell nuclear antigen antibody (PCNA; 1:200 dilution with 3% BSA; Sigma, P8825; this antibody is derived from the PC10 hybridoma created by the fusion of mouse myeloma cells and splenocytes from a BALB/c mouse immunized with PCNA–Protein A fusion protein; Gene ID: 18538) at 4 °C overnight. After 3 washes with 500 μL PBS, cells were incubated (room temperature, 1.5 h) with 200 μL goat anti-rabbit-FITC (1:250 dilution with 3% BSA; Sigma, F1262) and 200 μL anti-mouse-TRITC (1:250 dilution with 3% BSA; Sigma, T5393), respectively, and rinsed with PBS 3 times. Subsequently, each well was subjected to 200 μL DAPI (1:500 dilution with PBS; Sigma, D9542) treatment (room temperature, 15 min) and PBST washing once. Finally, the gold antifade reagent (Invitrogen, Carlsbad, CA, USA) was used for cell climbing slide mounting.

### 2.8. Cell Transfection

pCVpr plasmid DNA (1 μg) was transfected into ObSSCs in gelatin-coated 24-well plates at 60% confluence using the GeneJuice reagent (Novagen, Darmstadt, Germany) according to a previous description [21]. Briefly, cells were seeded; after growing to 60–70% confluence, cells were transfected with plasmids following the instructions of the GeneJuice reagent. After 6 h of incubation at 28 °C, 1 mL EMS4 was added, and the transfection efficiency was observed after culturing in the cell incubator at 28 °C for 24–48 h. Puromycin (10–15 μg/mL) was administered to cell cultures after 48 h for drug selection. A single cell with strong red fluorescent protein (RFP) expression was selected and cultured in a new 96-well, and then a monoclonal ObSSC line stably expressing RFP was obtained.

### 2.9. Induced Differentiation

RFP-positive ObSSCs were seeded on 24-well plates for suspension culture with all-trans retinoic acid (10 μM) to generate embryoid bodies (EBs), as previously described [22]. Following 7 days of suspension culture, single cells were separated from EBs and seeded on gelatin-coated 24-well plates to allow cells to adhere overnight.

### 2.10. Cell Transplantation

Cell transplantation was carried out essentially as reported [23,24]. Trypsinization resulted in a single RFP-positive ObSSC. Dechorionated zebrafish embryos (treated with 5 mg/mL pronase E at the blastula stage for 4 min) were collected and arranged in balanced salt saline (0.3% Instant Ocean Salt) with 2 ppm of methylene blue. Each midblastula recipient received 100–200 ObSSCs, which were injected into the deep cell layer. The embryos were cultured (28 °C) in zebrafish egg water (0.3% Instant Ocean Salt containing 2 ppm of methylene blue) and were periodically checked.

### 2.11. Coculture and Flow Cytometry

After trypsinization, the ESM4, gelatin-coated 24-well plates were seeded with 10^4^ RFP-positive ObSSCs and 10^4^
*O. bidens* testicular cells (ObCTs) at 4 passages. During two weeks of continuous induction, half of the medium was replaced every two days for coculturing of cells at 28 °C.

For flow cytometry, RFP-positive ObSSCs (p70) and ObCTs (p4) were collected by trypsinization. After fixing in 70% ethanol, cells underwent 15 min incubation (37 °C) in PBS supplemented with 200 μg/mL RNase A and 20 μg/mL propidium iodide for DNA staining. BD Accuri C6 flow cytometer and ModFitWinTrial software were used for analysis.

## 3. Results

### 3.1. Establishment of a Spermatogonial Stem Cell Line

The feeder-free culture system developed in medaka was successful for embryonic stem cells [25], SSCs [16], and haploid stem cell derivation [22], which was adopted for the establishment of an ObSSC line (Figure 1). Briefly, dissociated testicular cells from 6-month-old *O. bidens* were seeded into gelatin-coated 24-well plates in the culture medium (Figure 2A). A large number of testicular cells were attached to the cell plate (about 50–60%) on the third day, and many embryoid bodies (EBs) formed (Figure 2B). The number of spermatogonial stem cells around EBs was identified by immunofluorescence (Appendix A), and the cells were selected from the surrounding EBs for subculture.

Alkaline phosphatase (AP) staining is effective in identifying embryonic germ cells, embryonic stem cells, and other pluripotent stem cell types [26]. In medaka, AP is highly expressed in embryonic stem cells and SSCs [16,22,25]. In our study, when testicular cells from *O. bidens* were stably subcultured for 20 days (Appendix A), 60% of adherent cells were found to be strongly positive for AP staining, while the remaining adherent cells were negative for AP staining (Appendix A). With the passage, the number of AP-negative cells gradually decreased. A spermatogonial stem cell line was obtained from the testis of male *O. bidens*, which had been subcultured for more than 70 passages in 200 days without any senescence, termed *O. bidens* spermatogonial stem cells (ObSSCs). ObSSCs were smooth in outline and oval in shape with sparse cytoplasm and uniform morphology (Figure 2C), and they consistently maintained high AP activity (Figure 2D).

### 3.2. Characterization of the Spermatogonial Property of ObSSCs

The chromosomal morphology of ObSSCs is shown in (Figure 2E). Most cells exhibited a diploid karyotype of 76 chromosomes, which is the characteristic of this organism [27]. Flow cytometry analysis also revealed the diploid structure (see below). Moreover, a small number of haploid cells were also observed (Appendix A). These results indicated that ObSSCs exhibited genetic stability during long-term culture.

The identity of ObSSCs was further analyzed by using germ cell markers. *Dazl* is specifically expressed in the germ cells of medaka and *Coilia nasus* and is highly expressed in spermatogonia [28,29]. *Dnd* has been characterized as a germ cell marker in a variety of vertebrates. *Dnd* is expressed in the spermatogenic cells of the four-eyed sleeper and highly expressed in the spermatogonia of *Oryzias celebensis* and medaka [15,30]. *Vasa* showed high expression in spermatogonia and spermatocytes of medaka and swamp eel [14,31]. *Gfra1*, as a molecular marker of rodent SSCs, is also expressed in SSCs of Nile tilapia, medaka, swamp eel, and rainbow trout [14,32,33]. *Nanog* is a molecular marker of stem cells, which is abundantly expressed in SSCs of some fish such as medaka, zebrafish, and rainbow trout [34,35]. *Dmrt1* is often used as a somatic marker gene, as in the identification of large yellow croaker gonadal somatic cell lines and gonadal Sertoli cells of medaka [16,36]. According to the in situ hybridization expression pattern of *O. bidens* gonad sections, *dmrt1* is expressed in both germ cells and somatic cells (data not shown). Our results showed that the reproductive-specific genes *dazl*, *dnd*, *vasa*, and *gfra1* and stem cell marker gene *nanog* were highly expressed in ObSSCs, while the somatic marker gene *dmrt1* was relatively poorly expressed (Figure 3A).

ObSSCs showed a high proliferation rate. When the passage ratio was 1:2 or 1:3, the number of cells nearly doubled in 2 days, and the cells maintained rapid growth for more than 1 year. Through immunostaining, ObSSCs were characterized and mitotic activity was determined. The ObSSCs showed a strong proliferating cell nuclear antigen (PCNA) signal, and almost all cells exhibited positive expression for germ cell marker VASA (Figure 3B–G). After ObSSCs (at Passage 80) were subjected to 2 weeks of cryopreservation and recovered, most cells still had strong AP activity (Appendix A).

### 3.3. Differentiation Induction of ObSSCs In Vitro

To unambiguously identify the ObSSCs, we transfected the plasmid pCVpr into ObSSCs at Passage 40. A satisfactory efficiency (15–20%) was achieved with the GeneJuice reagent. The red fluorescent protein (RFP)-expressing ObSSCs were clonally expanded for 2 months (Figure 4A–C). The RFP-expressing ObSSCs still maintained the same high growth rate and high AP activity as untransfected ObSSCs (Appendix A). ObSSCs formed three-dimensional EBs in suspension culture. After continuous treatment with retinoic acid for one week, the types of terminally differentiated cells from EBs were identified, including neurons, astrocytes, and epithelial cells, suggesting the pluripotency ability of ObSSCs in vivo (Figure 4G–I).

### 3.4. Pluripotency of ObSSCs In Vivo

Furthermore, the in vivo pluripotency of ObSSCs was tested by cell-embryo transplantation. With RFP-expressing ObSSCs as transplant donors, 100–200 cells were transplanted into zebrafish blastocysts. A total of 80 zebrafish embryos were transferred, of which 11 were found to be positive (13.75%). Among the transfer-positive embryos, five embryos were able to develop normally, and the remaining embryos died due to excessive mechanical damage during transfer. To explore the development of ObSSCs in the zebrafish host, we observed the recipients for one week after transplantation. ObSSCs were not rejected in the zebrafish host but followed the embryonic development of zebrafish to form chimeras, showing normal proliferation and differentiation ability in zebrafish (Figure 5). Due to the different positions of the blastocyst transplanted, ObSSCs differentiated into a variety of tissues, such as vascular epithelial cells, skin cells, and muscle tissues (Appendix A). The results show that ObSSCs adopted the developmental program of the zebrafish host and proliferated and differentiated into various tissues from three germ layers in zebrafish.

### 3.5. Sperm Production of ObSSC In Vitro

SSCs reside in a specialized microenvironment in the testis of vertebrates and maintain spermatogenesis homeostasis by providing growth factors and cell-to-cell interactions [37,38]. To simulate the process of meiosis in vitro, we co-cultured ObSSCs with ObCTs (at Passage 4). We collected RFP-expressing ObSSCs and ObCTs separately for chromosome examination by flow cytometry. ObCTs contain three peaks of haploid, diploid, and tetraploid DNA content, indicating that the cells can enter and complete meiosis to produce haploid products (Figure 6O). ObSSCs exhibit two peaks of diploid and tetraploid DNA content (Figure 6P).

From Day 1 of coculture, we exposed cells to the EMS4 medium containing fetal bovine serum (FBS), basic fibroblast growth factor (bFGF), fish serum, and medaka embryo extract (MEM). It can be observed that RFP-expressing ObSSCs that grow uniformly before coculture are clustered under coculture with ObCTs (Figure 6A–D). After one week, ObSSCs and ObCT together formed EBs (Figure 6E–H). The volume of the EB increased after two weeks and did not increase significantly in the third week (Appendix A).

During spermatogenesis, four round spermatids from a single primary spermatocyte are connected into a tetrad through cytoplasmic bridges [39]. On Day 3 of coculture, a clustered tetrad consisting of four round cells from ObCT was observed in the medium (Figure 6I,J). On Day 14, RFP-expressing sperm cells were observed in the cultured medium, showing that ObSSCs were gradually induced into immature sperm cells (Figure 6K,L). After 16 days of coculture, it was observed that RFP-expressing ObSSCs were differentiated into more mature sperm with slight wobbling behavior in their tails; however, they could not swim normally (Figure 6M,N). Briefly, these findings suggested that, under suitable induction conditions, ObSSCs could enter the spermatogenesis stage to produce sperm cells.

## 4. Discussion

In this study, a normal SSC line was established for the first time in farmed fish, Chinese hook snout carp, and it was verified that the ObSSC line could produce sperm by coculture with testicular cells. ObSSCs were identified by various methods, including AP activity staining, identification of germ stem cell marker genes, cellular immunohistochemistry, and its potential spermatogenesis ability. ObSSCs underwent cryopreservation and resuscitation in vitro, followed by more than 2 years of subculture. ObSSCs had a stable growth rate and a uniform oval shape. The results confirmed that ObSSCs stably showed the characteristics of SSCs, indicating the successful establishment of the *O. bidens* spermatogonial stem cell line.

An appropriate culture system is a key to successfully establishing a spermatogonial stem cell line. A previous study reported that high-level FBS not only promoted the proliferation of germ cells in rainbow trout, but also promoted the rapid proliferation of somatic cells, which was not conducive to the continued growth of germ cells [40]. Zebrafish SSCs cultured with FBS were also unable to maintain their normal proliferation [41]. In salmon, FBS was replaced by bovine serum (BSA) and salmon serum to inhibit somatic cell growth and maintain the biological properties of cultured salmon SSCs in a short time [42]. The growth factors in medaka embryo extracts are also important for maintaining the properties of medaka embryonic stem cells and SSCs [16,20]. The medium used in this study was partially modified on the basis of the EMS4 medium. We mixed FBS with sea bass serum in an appropriate ratio and added medaka embryo extract and bFGF. Interestingly, although medaka and *O. bidens* are cross-order species, the requirement for stable growth of ObSSCs under feeder-free conditions is very similar to that of medaka. Showing that the growth factors in medaka embryos still play an important role in maintaining ObSSCs’ biological characteristics.

Recent studies have shown that SSCs have the potential to differentiate into various cells after induction. For example, mouse male SSCs can be transformed into oocyte-like cells in culture [43]. Embryonic stem-like cells from mouse SSCs can be transformed into retinal ganglion cells with morphological and functional characteristics [44]. Human SSCs can generate biologically functional neurons in vitro [45]. Notably, we found that retinoic acid induced ObSSCs to differentiate into nerve cells, epithelial cells, and other somatic cells. In medaka, a previous study reported that the spermatogonial cell line (SG3) differentiates into four different ectodermal and mesodermal somatic cell types in vitro [46]. After microinjection of ObSSCs into zebrafish embryos at the blastula stage, ObSSCs proliferated and differentiated with the development of zebrafish embryos to form chimeras, and the differentiative capacity of donor cells into multiple tissues in zebrafish embryos represented the most convincing demonstration of their pluripotency both in vitro and in vivo.

Spermatogenesis goes through three main stages: self-renewal of spermatogonia, meiosis, and sperm production. During meiosis, round sperm cells are formed to further generate mature sperm. Previous studies have shown that in lower vertebrates, spermatogenesis can be performed in vitro. In medaka, coculture of the SG3 cell line with gonadal somatic cells from rainbow trout or *Coilia nasus* [47] can induce sperm in vitro [16]. Recently, the whole spermatogenesis process of mice was reproduced in vitro by combining IPS induction technology and 3D culture technology, and the generated organoids formed a structure similar to seminiferous tubules, in which spermatogenesis was performed and functional sperm was obtained [10,11]. In the current study, flow cytometry results revealed that long-term-cultured ObSSCs could not spontaneously enter the spermatogenesis program. Interestingly, coculture by simulating the testicular environment in vivo not only facilitated the production of sperm by ObSSCs, but also induced them to re-form larger gonadal organoids. Whether spermatogenesis can be performed inside the organoids generated in this experiment requires further research.

## 5. Conclusions

In conclusion, SSCs have unique advantages over other types of stem cells, which are excellent models for stem cell biology. Especially in recent years, the widespread application of SSCs in fish germ cell transplantation has gradually become an important aspect of fish reproductive development. In Chinese rare minnow (*Gobiocypris rarus*), gene-edited sperm derived from different subfamily species were obtained by transplanting SSCs into zebrafish [48]. The establishment of the *O. bidens* SSC line contributes to exploring the potential and regulatory mechanism of fish SSC differentiation and also lays a foundation for further research on fish germ cell transplantation, which is of great importance to the protection of the endangered fish species and the optimization of fish resources and confers novel insight into fish breeding. Further studies are needed to determine whether the SSC of *O. bidens* can differentiate into functional gametes and produce offspring of *O. bidens* upon transplantation to surrogate fish.

## Figures and Tables

**Figure 1 biology-11-01069-f001:**
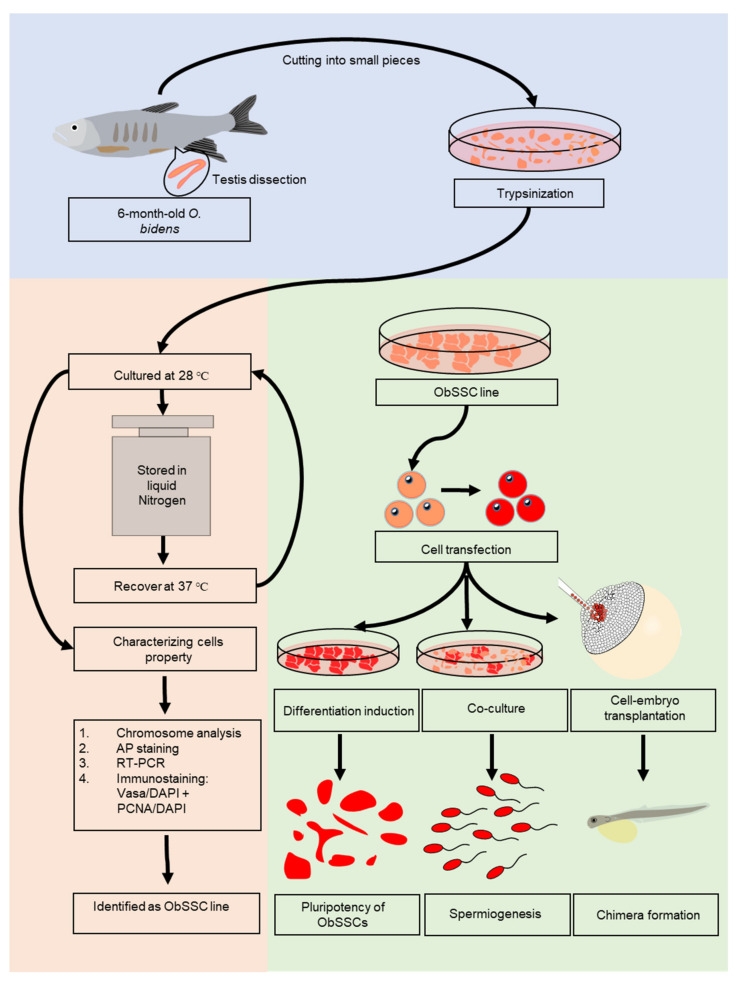
Flowchart of experiments. In total, seven 6-month-old male adult fish with a length of about 10 cm were used. Live fish were placed on ice, and there was no response to mechanical stimulation. Testicular tissues were dissected and washed two times with phosphate-buffered saline (PBS) containing 1% penicillin/streptomycin. All testes were minced and dissociated by trypsin and collagenase. A stable ObSSC line was obtained by appropriate culturing and then cryopreserved. We verified that these cells were SSCs by chromosomal analysis, RT-PCR and immunofluorescence. ObSSCs were identified in vitro and in vivo with pluripotent stem cell properties. Importantly, ObSSCs were induced to sperm by coculture with testis cells.

**Figure 2 biology-11-01069-f002:**
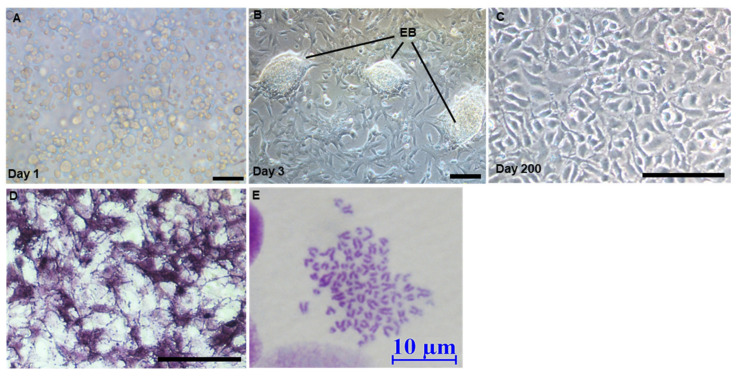
Derivation of ObSSCs. (**A**) Testicular cells after enzymatic dissociation. (**B**) *O. bidens* primary testis cells were cultured on the 3rd day. (**C**) Morphology of ObSSCs at Passage 78 and Day 200. (**D**) Alkaline phosphatase staining. EB, embryoid bodies. (**E**) Diploid metaphase of ObSSCs. (Bars = 100 μm unless indicated).

**Figure 3 biology-11-01069-f003:**
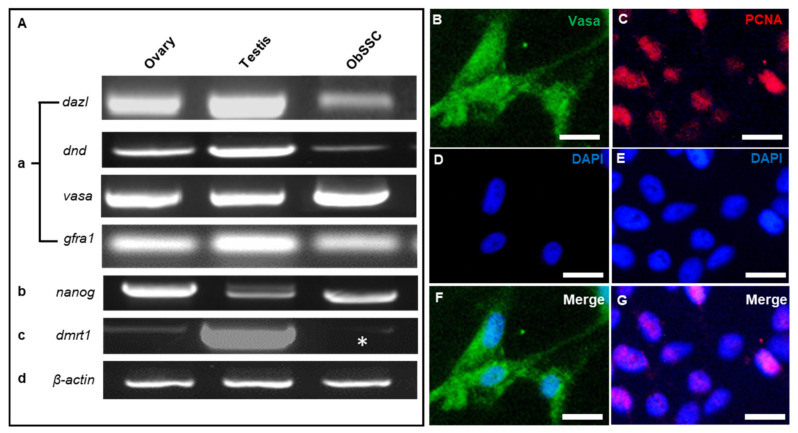
PCR and immunofluorescence analysis of ObSSCs. (**A**) Expression of germ cell markers by RT-PCR of total RNA from ObSSCs and adult gonads with primers for *dazl*, *dnd*, *vasa*, and *gfra1*. (**B**) Expression of stem cell marker by RT-PCR with primers for *nanog*. (**C**) Expression of Sertoli cell marker by RT-PCR with primers for *dmrt1* (asterisk). (**D**) *Actin* expression was determined for calibration. (**B**–**G**) Immunofluorescence of Vasa (green) and PCNA (red) in ObSSCs. (Bars = 20 µm).

**Figure 4 biology-11-01069-f004:**
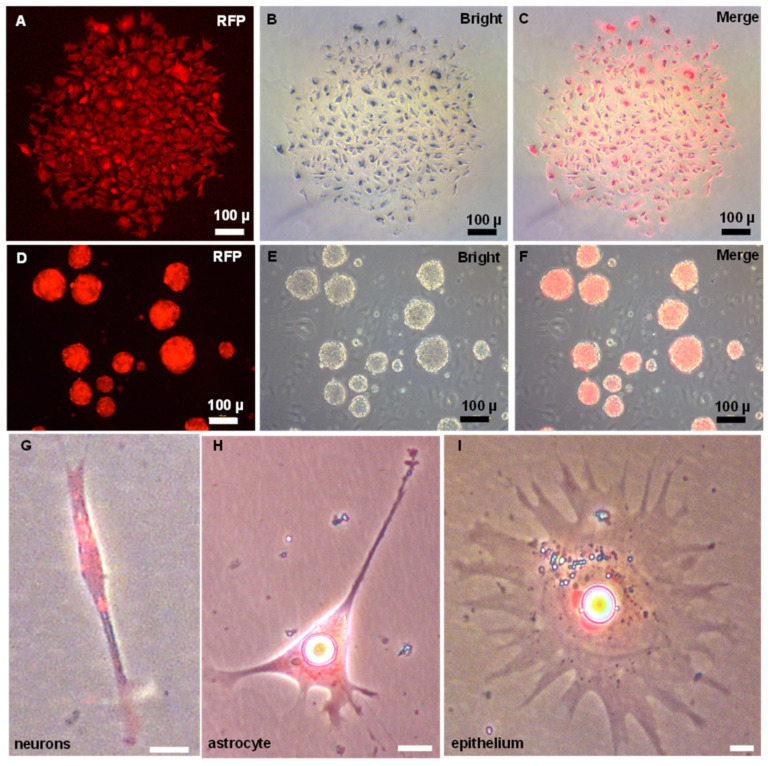
Pluripotency of ObSSCs in vitro. (**A**–**C**) Single colony of ObSSCs expressed with RFP by transfection with pCVpr at Passage 40. (**D**–**F**) Embryoid bodies formed by suspension culture of ObSSCs labeled with RFP. (**G**–**I**) The types of terminally differentiated cells from EBs. (Bars = 10 µm unless indicated).

**Figure 5 biology-11-01069-f005:**
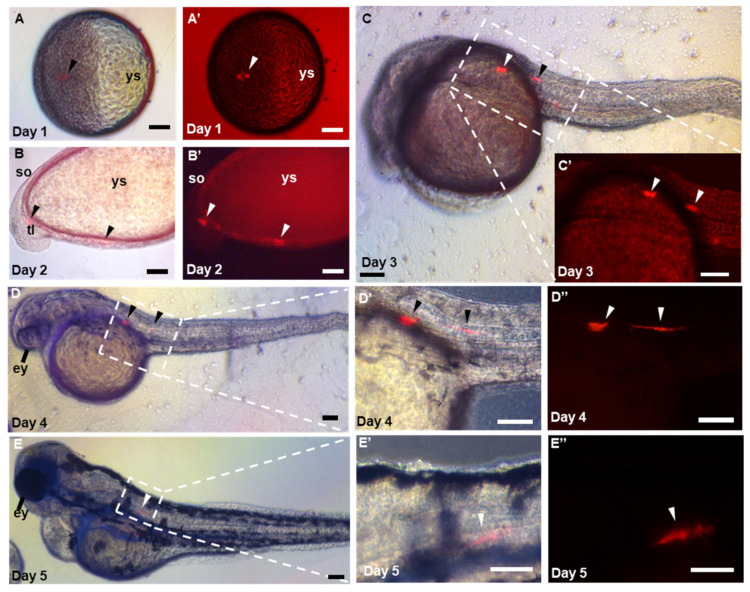
Pluripotency of ObSSCs in vivo. (**A**–**E**) Different stages of embryo, in which the chimeras were observed by microscopy at 1 to 5 days post-fertilization. (**A**–**D**,**D′**,**E**,**E′**) Merge of bright and fluorescent fields. (**A′**–**C′**,**D″**,**E″**) Red fluorescent micrograph. ys, yolk sac; ey, eye; so; somite; tl; tail; ht; heart. (Scale bars, 100 µm).

**Figure 6 biology-11-01069-f006:**
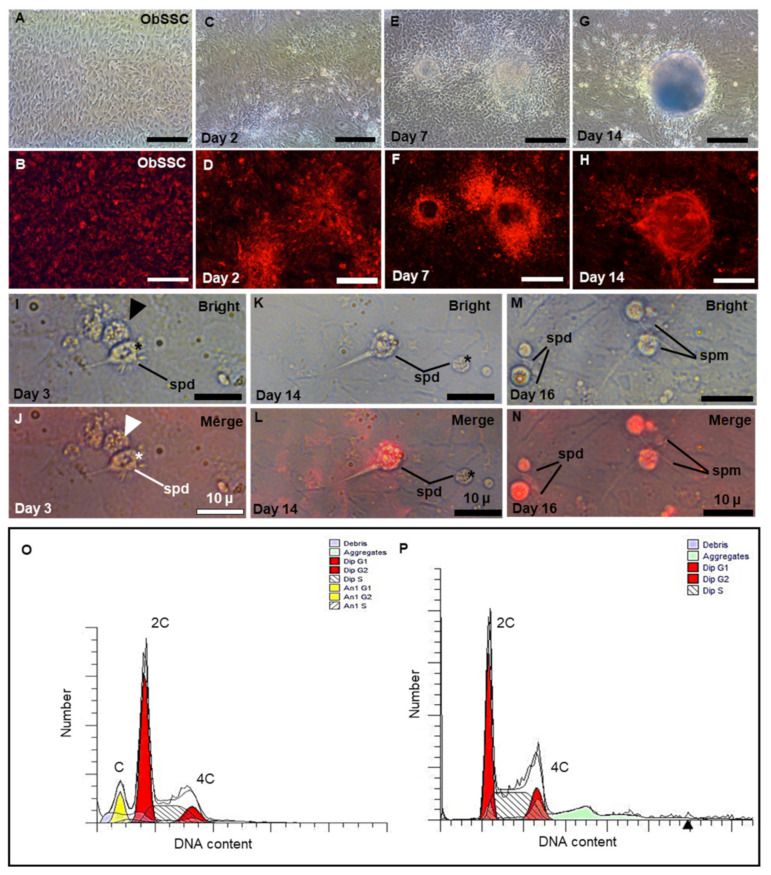
Sperm production of ObSSCs in vivo. (**A**,**B**) The RFP-expressing ObSSCs were evenly distributed in the culture wells. (**C**–**H**) Coculture of *O. bidens* primary testicular cells and ObSSCs at Passage 70 were maintained at confluence without subculture for 2 weeks. (**I**–**N**) After coculture of *O. bidens* primary testicular cells and ObSSCs, sperm production can be observed in the culture medium. (**I**,**J**) Sperm from *O. bidens* primary testis cells. A cluster of four sperm from a tetrad is indicated by an arrow. (**K**–**N**) RFP-positive, spermatid, and sperm with a thin and long tail. (**O**) Primary testicular cells showing a prominent haploid peak in addition to diploid and tetraploid peaks. (**P**) ObSSCs at Passage 70 were maintained under normal culture conditions for undifferentiated proliferation, showing diploid and tetraploid peaks. c, 2c, and 4c, haploid, diploid, and tetraploid DNA contents, respectively. (Scale bars, 100 µm in (**A**–**H**); 10 µm in (**I**–**N**)).

**Table 1 biology-11-01069-t001:** Primers used for reverse transcription polymerase chain reactions (RT-PCR).

Gene	Primer Sequence	
Name	Forward Primer	Reverse Primer
*dazl*	ATGAAGGTGGAAGAGAACGAGAT	GGAGATGGCAGTGAACGAGAA
*dnd*	TGGTTCGCAAGAGCACAGA	TGTTCGCCTCGCAGATCAT
*vasa*	TGGAAGTGAGCGGCAGCAATG	CCACCACACCAACAGCAAGGAA
*gfra1*	GTAGTGCGTCGGACTGAGT	ATGTCGCCTGCTGTTGGA
*nanog*	AGTGATGGCAGATTGGACGATA	CGGTTGAGCGGTGTAATAGC
*dmrt1*	GTGCCAGATGTCGGAACCA	AACCTCGGATCTTGCTTGCT
*β-actin*	TTCAACAGCCCTGCCATGTAC	CCTCCAATCCAGACAGAGTATT

## Data Availability

Not Applicable.

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
