# Peer review of "Generation of a Normal Long-Term-Cultured Chinese Hook Snout Carp Spermatogonial Stem Cell Line Capable of Sperm Production In Vitro"

_biology, 2022, doi:10.3390/biology11071069_

Round 1

Reviewer 1 Report

The manuscript “Generation of a normal long-term cultured Chinese hook snout carp spermatogonial stem cell line capable of sperm production in vitro” describes a successful in production spermatogonial cell line of Chinese hook snout carp. I read the manuscript carefully. The success of technique is unquestionable and promising; however, some points need to be explained and text must be improved.  There are several comments listed below, please check and revise them for publication. 

Line45: “provides new ideas for protecting endangered fish” In my opinion, this phase is overclaim since the Iwasaki-Takahashi et. al., 2020 published the article “Production of functional eggs and sperm from in vitro-expanded type A spermatogonia in rainbow trout”

Line57 & 60: What is GSCs stand for? Germline stem cell or germ line-competent stem cells?

Where did you get the fish from?

Line99: Describe the condition for the automated breeding system.

Please provide the company names for all chemical.

Line107: Is it the commercial fish serum? Or you prepare it? Please provide the detail.

Line106: You used both “ml” and “mL”. Please check the consistency through the manuscript.

Line122: How did you identify obSSC?

Line 126 & 127: You used both “h” and “hour”. Please check the consistency through the manuscript.

Line127: Describe the protocol of colchicine treatment briefly.

Line130: Please convert to final concentration.

Line131: Did you mean microcentrifuge tube for the “Eppendorf”?

Line133: What kind of fixative did you use?

Line139: 20% DMSO is very high concentration for slow freezing cryopreservation.  Did you observe the survival rate of the cells after thawing?

Line139: How many cell volumes did you put into 1.5 ml cryogenic vials?

Line152: “to make cDNA from 1 g total RNA” Is it mistake? Why did you use very high amount of total RNA for cDNA synthesis???

Line155: “95°C for 5 min, 95°C for 20s, 58°C for 20s, 72°C for 30s-60s and extension 155 at 72°C for 10min” Please rewrite to be a sentence.

Line161-172: Please rewrite this paragraph. The most of them are phrase!!!

Line161: “Transfer the slides to a new 24- 161 well plate with the cells facing up after washing with PBS” What did you mean?

What solution did you used for anti-body dilution?

Line176: “described previously” Describe the protocol briefly.

Line178: “Pick a red fluorescent cell and place it in a 178 new well, then culture and expand the monoclonal cell line to generate an ObSSC line that 179 stably expresses red fluorescent protein” rewrite to be sentences.

Line197: “T” to “t”

Line201: What did you mean for “respectively”?

Line204: analyzing

I could not find the explanation of figure1 in the text.

Please provide a method for karyotype analysis

Figure3: Please provide scale bars in all Figure 3B-G.

Figure3: Why cells in PCNA group were denser than vasa group? Please explain.

Line281: “The RFP-expressing ObSSCs still main- 281 tained a faster growth rate and higher AP activity” What did you compare RFP-expressing ObSSCs with?

Line282: Which figure did you show the AP activity of ObSSCs?

Figure4: What is the difference between 4A, 4B, and 4C?

Figure4: According to the scale bars in the Fig 4A and 4B, are they the same magnification? It looks like different magnification for me.

Figure4: What is the difference between 4G, 4J, and 4K?

Figure4: What is the difference between 4H, and 4M?

 Figure4: What is the difference between 4I, and 4L?

How was the efficiency for plasmid pCVpr into ObSSCs transfection?

Figure5: Did you observe the control recipients (non-transplanted recipients)? It would be better to provide non-transplanted recipient picture as a control in this figure.

How many recipients did you observe the red fluorescent? Please provide the efficiency.

Line317: “Under normal culture conditions” What is it mean?

Line322: “The volume of the EB 322 increased after two weeks but did not increase significantly until the third week (Fig. S3).” You should provide the image of 1st week to compare to 3rd week.

Line330: “slight wobbling behavior in their tails, and 11 of 15 their tails had a slight wobbling behavior” It’s repeated!!! Please rewrite.

Please refer Figure 6A and 6B in the text.

Figure6: What is the difference between Figure 6I, K, M and 6J, L, N. Please provide more details in the figure caption.

Supplementary Figure1: “(A-F) 411 Lower magnification. (D-F) Higher magnification.” Confused!!

Supplementary Figure2: What is the difference between 2A, and 2B? Please provide more details in the figure caption. 

Author Response

Thank you for giving me opportunity to submit our revise manuscript entitle “Generation of a normal long-term cultured Chinese hook snout carp spermatogonial stem cell line capable of sperm production in vitro”. We appreciate the time and effort you and the reviewers put into providing insightful feedback on our manuscript. We were able to incorporate changes that reflected the majority of the reviewers' suggestions. The changes have been highlighted in the manuscript.

Here is a point-by-point response to the reviewers’ comments and concerns;

Reviewer 1:

The manuscript “Generation of a normal long-term cultured Chinese hook snout carp spermatogonial stem cell line capable of sperm production in vitro” describes a successful in production spermatogonial cell line of Chinese hook snout carp. I read the manuscript carefully. The success of technique is unquestionable and promising; however, some points need to be explained and text must be improved.  There are several comments listed below, please check and revise them for publication.

  1. Line45: “provides new ideas for protecting endangered fish” In my opinion, this phase is overclaim since the Iwasaki-Takahashi et. al., 2020 published the article “Production of functional eggs and sperm from in vitro-expanded type A spermatogonia in rainbow trout”.

Response: Thanks for your suggestion. The article published by Iwasaki-Takahashi in 2020 is to obtain spermatogonial stem cells in a short time. However, our experiment is to obtain spermatogonial stem cells that can be cultured in vitro for a long time, and a large number of spermatogonial stem cells can be obtained through cell culture. And we have adjusted the sentences in the abstract section, "Our research helps to explore the potential and regulation mechanism of fish SSC differentiation, provides new ideas for protecting endangered fish, solves the problem of fish germplasm resource degradation, and lays a foundation for further research on fish germ cell transplantation" changed to "Our research helps to explore the potential and regulation mechanism of fish SSC differentiation, spermatogenesis in vitro, and provides a new way for solving the problem of fish germplasm resource degradation, and lays a foundation for further research on fish germ cell transplantation" in lines 44-47.

  1. Line57 & 60: What is? Germline stem cell or germ line-competent stem cells?

Response: We have adjusted the GSCs stand for “Germline stem cells” in line 60.

  1. Where did you get the fish from?

Response: We have added fish information “The fry fish were obtained in Jinhua Hengyuan Agricultural Science and Technol-ogy Co., Ltd. (Zhejiang, China) and were transported to Shanghai” in lines 100-101.

  1. Line99: Describe the condition for the automated breeding system.

Response: We have added conditions for the automated breeding system in lines 101-103 “And they were raised (26 °C) in the automated breeding with a 14 h:10 h light/dark cycle in the Shanghai Ocean University Opsariichthys bidens Center (Shanghai, China)”.

  1. Please provide the company names for all chemical.

Response: We have added company names for all chemical in the article.

  1. Line107: Is it the commercial fish serum? Or you prepare it? Please provide the detail.

Response: The fish serum is prepared by ourselves. We have added references to fish serum extraction methods in line 112.

  1. Line106: You used both “ml” and “mL”. Please check the consistency through the manuscript.

Response: We have adjusted “ml” to “mL” in the manuscript.

  1. Line122: How did you identify obSSC?

Response: The ObSSCs were observed with uniform morphology and rounded cell edges under a microscope. Karyotyping identified the cell line as having a diploid karyotype, carrying 38 pairs of chromosomes. Through RT-PCR results, the ObSSC line simultaneously expressed dazl, dnd, nanog, vasagenes related to germinal stem cells, and it was detected to have strong alkaline phosphatase activity. Vasa and PCNA signals were strong in cellular immunofluorescence. The cells were identified after about 200 days of long-term culture, and the cell line was still able to maintain its stable biological properties. It shows that we have successfully cultivated a spermatogonial stem cell line that can be cultured in vitro for a long time.

And we have adjusted the sentences in lines 125-129 " Briefly, dissociated cells were cultured in EMS2 medium for several weeks, passaged to passage 15 and a stable ObSSC line was obtained. From passage 15 of the culture, Ob-SSCs were cultured in fresh ESM4 medium" changed to " Briefly, dissociated cells were cultured in EMS2 medium for several weeks, passaged to passage 15 and stably proliferating ObSSCs were obtained. From passage 15 of the cul-ture, ObSSCs were cultured in fresh ESM4 medium, the cells were passaged to 50 pas-sages and named the ObSSC line, which has more than 100 passages in 2 years of culture ".

  1. Line 126 & 127: You used both “h” and “hour”. Please check the consistency through the manuscript.

Response: We have corrected this issue throughout the manuscript.

  1. Line127: Describe the protocol of colchicine treatment briefly.

Response: We have describe the protocol of colchicine treatment briefly as follows: “ Cells with active growth were cultured (4 h, 28°C) in colchicine solution (20 μg/mL) to accumulate metaphases” in lines 132-134.

  1. Line130: Please convert to final concentration.

Response: We have converted to the final concentration as follows: “Cells were put in microcentrifuge tubes (1.5 mL) and gradually added with hypotonic solution (1 mL, 2.98 mg/mL KCl) for 30-min incubation for cell swelling” in lines 135-137.

  1. Line131: Did you mean microcentrifuge tube for the “Eppendorf”?

Response: Yes. We have changed “Eppendorf tubes” to “microcentrifuge tube” in line 136.

  1. Line133: What kind of fixative did you use?

Response: Thanks for your reminding us this important point. We have detailed the fixative as follows: “After that, the hypotonic treatment was terminated by adding fresh fixative (0.2 mL, 3:1 methanol:acetic acid)” in lines 137-139.

  1. Line139: 20% DMSO is very high concentration for slow freezing cryopreservation. Did you observe the survival rate of the cells after thawing?

Response: We have changed “30% FBS and 20% dimethyl sulfoxide” to “15% FBS and 10% dimethyl sulfoxide” in line 145. Here we are missing a step. Before cryopreserving cells, we will dilute 1:1 between cryopreservation medium and EMS4 medium. So the final concentrations of FBS and dimethyl sulfoxideshould be 15% and 10%, respectively. The survival rate of the cells after thawing is about 80%-90% (cells cryopreservation for half a year).

  1. Line139: How many cell volumes did you put into 1.5 ml cryogenic vials?

Response: We have changed “The ObSSCs were resuspended at a density of 2 X 105 cells/mL in a medium contain-ing 15% FBS and 10% dimethyl sulfoxide” to “The 4 X 105 ObSSCs were resuspended in 1.5 mL medium containing 15% FBS and 10% dimethyl sulfoxide” in lines 145-146.

  1. Line152: “to make cDNA from 1 g total RNA” Is it mistake? Why did you use very high amount of total RNA for cDNA synthesis???

Response: Should be “1 µg”, we have corrected it in line 158.

  1. Line155: “95°C for 5 min, 95°C for 20s, 58°C for 20s, 72°C for 30s-60s and extension 155 at 72°C for 10min” Please rewrite to be a sentence.

Response: We have changed “RT-PCR was performed for 35 cycles in a 20 μL system. 95°C for 5 min, 95°C for 20s, 58°C for 20s, 72°C for 30s-60s and extension at 72°C for 10min” to “PCR amplification conditions were: 95 °C for 5 min, and then 35 cycles of 95 °C for 20 s, 58 °C for 20 s, and 72°C for 30 s-60 s in a 20 μL system; followed by 72 °C for 10 min” in lines 160-162.

  1. Line161-172: Please rewrite this paragraph. The most of them are phrase!!!

Response: We have rewrited this paragraph in lines 167-185 “After 2 PBS washes, cells were fixed (10 min) in 4% paraformaldehyde. Following PBS washing, the cell climbing slides were transferred to a new 24-well plate, and the cell-covered side was facing up. Cell were subjected to 10 min 0.1 % Triton-X 100 (200 μL) treatment and then 2 PBS washes. Next, cells were blocked (30 min) with 3% bovine se-rum albumin (BSA, 200 μL, Sigma), and washed with 500 μL of PBS once. Next, the cells in one well were incubated with 200 μL of Vasa antibody (1:200 dilution with 3% BSA; Invitrogen, PA5-30749; this antibody corresponds to a recombinant fragment of human DDX4, GeneID:54514) while the cells in another well were added with 200 μL of proliferating cell nuclear antigen anti-body (PCNA; 1:200 dilution with 3% BSA;Sigma,P8825;derived from the PC 10 hybridoma generated by the fusion of mouse myeloma cells and splenocytes from a BALB/c mouse immunised with PCNA-Protein A fusion protein, GeneID:18538) at 4°C overnight. After 3 washes with 500 μL of PBS, cells were incubated (room tem-perature, 1.5 h) with 200 μL of goat anti-rabbit-FITC (1:250 dilution with 3% BSA; Sigma, F1262) and 200 μL of anti-mouse-TRITC (1:250 dilution with 3% BSA; Sigma, T5393) respectively, and rinsed with PBS 3 times. Subsequently, each well was subject-ed to 200 μL of DAPI (1:500 dilution with PBS; Sigma, D9542) treatment (room tem-perature, 15 min) and PBST washing once. Finally, the gold anti-fade reagent (Invitro-gen, Carlsbad, CA) was used for cell climbing slide mounting”.

  1. Line161: “Transfer the slides to a new 24-well plate with the cells facing up after washing with PBS” What did you mean?

Response: Most of the cells will only cover one side of the cell climbing slides, to prevent the cells from falling off and to have sufficient contact with the reagents, it is necessary to place the cell-covered side up. And we have changed “Transfer the slides to a new 24-well plate with the cells facing up after washing with PBS” to “Following PBS washing, the cell climbing slides were transferred to a new 24-well plate, and the cell-covered side was facing up” in line 168-169.

  1. What solution did you used for anti-body dilution?

Response: We have added “dilution with 3% BSA” in Section 2.7.

  1. Line176: “described previously” Describe the protocol briefly.

Response: We have added “Briefly, cells were seeded; after cell growing to 60%-70% confluence, cells were trans-fected following the instructions of the GeneJuice reagent. After 6 hours of incubation at 28 °C, 1 mL of EMS4 was added, and the transfection efficiency was observed after culturing in the cell incubator at 28 °C for 24 h-48 h” in lines 190-193.

  1. Line178: “Pick a red fluorescent cell and place it in a 178 new well, then culture and expand the monoclonal cell line to generate an ObSSC line that 179 stably expresses red fluorescent protein” rewrite to be sentences.

Response: We have changed “Pick a red fluorescent cell and place it in a new well, then culture and expand the monoclonal cell line to generate an ObSSC line that stably expresses red fluorescent protein” to “A single cell with strong red fluorescent protein (RFP) expression was selected and cul-tured in new 96-well, and then a monoclonal ObSSC line stably expressing RFP were obtained” in lines 194-196.

  1. Line197: “T” to “t”

Response: We have changed “T” to “t”.

  1. Line201: What did you mean for “respectively”?

Response: We removed the "respectively".

  1. Line204: analyzing

Response: We have corrected “analyzed” to “BD Accuri C6 flow cytometer and ModFitWinTrial software were used for analysis” in lines 219-220.

  1. I could not find the explanation of figure1 in the text.

Response: Figure1 is the flowchart, we have added it in line 222.

  1. Please provide a method for karyotype analysis

Response: We have added a method for karyotype analysis in section 2.4 as follows: “Under slides were photographed and examined through Nikon Eclipse 80I fluorescence microscope”.

  1. Figure3: Please provide scale bars in all Figure 3B-G.

Response: We have added scale bars in all Figure 3B-G.

  1. Figure3: Why cells in PCNA group were denser than vasa group? Please explain.

Response: Immunofluorescence of PCNA group and Vasa group was performed in two different cell wells, and the uneven distribution of cells in the cell wells resulted in different local densities.

  1. Line281: “The RFP-expressing ObSSCs still maintained a faster growth rate and higher AP activity” What did you compare RFP-expressing ObSSCs with?

Response: This sentence means that RFP-expressing ObSSCs did not change their biological characteristics after transfection. We have changed “The RFP-expressing ObSSCs still maintained a faster growth rate and higher AP activity” to “The RFP-expressing ObSSCs still maintained the same high growth rate and high AP activity as untransfected ObSSCs” in line 296-297.

  1. Line282: Which figure did you show the AP activity of ObSSCs?

Response: Due to a large number of pictures in the article, we did not put a figure showing the AP activity of RFP-expressing ObSSCs. We added this figure to Figure 2F in Supplementary Material.

  1. Figure4: What is the difference between 4A, 4B, and 4C?

Response: Fig 4A and 4B are taken in the same field of view under different light fields, and Fig 4C is the merged picture of Fig 4A and 4B. We have added annotations to the Fig 4A-4F.

  1. Figure4: According to the scale bars in the Fig 4A and 4B, are they the same magnification? It looks like different magnification for me.

Response: The magnification is the same for Figures 4A and 4B. It may be caused by unclear cell edges under brightfield observation.

  1. Figure4: What is the difference between 4G, 4J, and 4K?

Response: We have removed Fig 4J-4M and only show one of each type.

  1. Figure4: What is the difference between 4H, and 4M?

Response: As shown above.

  1. Figure4: What is the difference between 4I, and 4L?

Response: As shown above.

  1. How was the efficiency for plasmid pCVpr into ObSSCs transfection?

Response: We have added “A satisfactorily efficiency (15%-20%) was achieved with Genejuice reagent” in lines 294.

  1. Figure5: Did you observe the control recipients (non-transplanted recipients)? It would be better to provide non-transplanted recipient picture as a control in this figure.

Response: We have observed control recipients (non-transfer recipients). In fact we have selected embryos that have successfully transferred from a large number of transferred embryos for follow-up observation. Zebrafish embryos do not have autofluorescence in the early stages of development, so it is obvious under a fluorescence microscope whether the cells have successfully transferred into the recipient. We did not follow up on the control group because we were more concerned with the development of ObSSCs in zebrafish embryos.

  1. How many recipients did you observe the red fluorescent? Please provide the efficiency.

Response: We have added “A total of 80 zebrafish embryos were transferred, of which 11 were found to be positive (13.75%). Among the transfer-positive embryos, 5 embryos were able to develop normally, and the remaining embryos died due to excessive mechanical damage during transfer” in line 310-313.

  1. Line317: “Under normal culture conditions” What is it mean?

Response: “Under normal culture conditions” means ObSSC cultured at 28°C in vitro. This sentence is redundant and we have deleted it.

  1. Line322: “The volume of the EB 322 increased after two weeks but did not increase significantly until the third week (Fig. S3).” You should provide the image of 1st week to compare to 3rd week.

Response: We have changed “but” to “and” and corrected “until” to “in” in line 341. The image in the first weeks of EBs growth is in Fig 6C, and the image in the third week is in Fig S3.

  1. Line330: “slight wobbling behavior in their tails, and 11 of 15 their tails had a slight wobbling behavior” It’s repeated!!! Please rewrite.

Response: We have adjusted the sentences “After 16 days of co-culture, it was observed that RFP-expressing ObSSCs were differentiated into more mature sperm with slight wobbling behavior in their tails, however, they could not swim normally” in line 349-351.

  1. Please refer Figure 6A and 6B in the text.

Response: We have adjusted the sentences in line 338-339 “It can be observed that RFP-expressing ObSSCs that grow uniformly before co-culture are clustered under co-culture with ObCTs (Fig. 6A-D)”.

  1. Figure6: What is the difference between Figure 6I, K, M and 6J, L, N. Please provide more details in the figure caption.

Response: We have added “(I-N) After co-culture of O. bidens primary testicular cells and ObSSCs, sperm production can be observed in the culture medium” in line 357-359.

  1. Supplementary Figure1: “(A-F) 411 Lower magnification. (D-F) Higher magnification.” Confused!!

Response: We have changed “(A-F) Lower magnification. (D-F) Higher magnification” to “(A-F) View under the low magnification. (D-F) View under the high magnification” in lines 436-437.

  1. Supplementary Figure2: What is the difference between 2A, and 2B? Please provide more details in the figure caption.

Response: We have added “Brightfield observation and alkaline phosphatase staining analysis of ObSSC cells at passage 6” in line 439-440.

Reviewer 2 Report

 Dear authors, thanks for the opportunity to review the manuscript ‘Generation of a normal long-term cultured Chinese hook snout carp spermatogonial stem cell line capable of sperm production in vitro’ (biology-1806858’. This work aimed to demonstrate culture, cryogenic storage, and differentiation of spermatogonia cell line (SSC) from Chinese hook snout carp (Opsariichthys bidens) (ObSSC). During more than 2 years of culture, ObSSC had a diploid karyotype and stable growth, with the typic gene expression patterns of SSC. Under in vitro culture, ObSSC could be differentiated into sperm and other different types of somatic cells. In vivo, ObSSC could differentiate into different cells of three germ layers upon being transplanted into zebrafish embryos. This is an important topic of research to explore germ cell manipulation techniques to preserve genetic resources for aquatic animals. The methods are sound, the results are well presented, and the discussion explains the important significance and implication of the work. I really enjoy the reading. Although no fully functional sperm were produced, this work provided a critical foundation for further exploration. My comments below are minor to help improve the quality of the reading. Thanks!

(1) List major objectives to be achieved in the Introduction.

(2) Line 139: ‘put’ can be replaced with ‘transferred’.

(3) Lines 139-140: Please provide manufacturers’ information for the ‘cryogenic box’ and ‘cooling box’.

(4) Line 140: Incubated at ‘80 C’? Are you sure about it?

(5) Lines 178-180: Please revise the grammar of this sentence.

(6) Line 211: Remove the space between ‘gelatin’ and ‘coated’.

(7) Figure 2: Please adjust the order of images. It is odd that the ‘E’ is next to ‘B’.

Author Response

(1) List major objectives to be achieved in the Introduction.

Response: We have adjusted the sentencesIn the present study, we successfully established a long-term cultured spermatogonia cell line capable of sperm production from the adult testis of O. bidens in vitro. And O. bidens SSCs can be induced to differentiate into a variety of somatic cells in vitro and in vivo. This study helps to provide a new model for the study of fish SSCs and lays the foundation for further studies on the reproduction of O. bidens. It also provides a donor for further research on germ cell transplantationin line88-94.

(2) Line 139: ‘put’ can be replaced with ‘transferred’.

Response: We have changed “put” to “transferred” in line 148.

(3) Lines 139-140: Please provide manufacturers’ information for the ‘cryogenic box’ and ‘cooling box’.

Response: We have added manufacturers’ information for the ‘cryogenic box’ and ‘cooling box’ in line 148-149.

(4) Line 140: Incubated at ‘80 C’? Are you sure about it?

Response: We have corrected “80 °C” to “-80 °C” in line 147.

(5) Lines 178-180: Please revise the grammar of this sentence.

Response: We have changed “Pick a red fluorescent cell and place it in a new well, then culture and expand the monoclonal cell line to generate an ObSSC line that stably expresses red fluorescent protein” to “A single cell with strong red fluorescent protein (RFP) expression were selected and cultured in new 96-well, and then a monoclonal ObSSC line stably expressing RFP were obtained” in line 194-196.

(6) Line 211: Remove the space between ‘gelatin’ and ‘coated’.

Response: We have removed the space.

(7) Figure 2: Please adjust the order of images. It is odd that the ‘E’ is next to ‘B’.

Response: We have adjusted the order of images in Fig 2.

There are some precautions when revising your

manuscript(biology-1806858), please pay attention:

  1. Please provide original Protein Data Bank figure of Figure 3.

Response: We have provided the original data of proteins in Figure 3: Vasa antibody (1:200 dilution with 3% BSA; Invitrogen, PA5-30749; this antibody corresponds to a recombinant fragment of human DDX4, GeneID:54514) and proliferating cell nuclear antigen antibody (PCNA; 1:200 dilution with 3% BSA; Sigma, P8825; this antibody is derived from the PC 10 hybridoma created by the fusion of mouse myeloma cells and splenocytes from a BALB/c mouse immunized with PCNA-Protein A fusion protein, GeneID:18538).

  1. Please reduce self-citation rate to below 15% (now 17%), if it is

still higher than 15% after revision, it will not process.

(currently: 5073, 1 table & 6 fig).

Response: We have reduced the self-citation rate by removing 5 our own papers as listed as below:

  • Hong, Y.; Schartl, M. Isolation and differentiation of medaka embryonic stem cells. Methods Mol Biol. 2006, 329, 3-16.
  • Xu, H.; Li, M.; Gui, J.; Hong, Y. Fish germ cells. Sci China Life Sci. 2010, 53, 435-446.
  • Zhu, T.; Gui, L.; Zhu, Y.; Li, Y.; Li, M. Dnd is required for primordial germ cell specification in Oryzias celebensis. Gene. 2018, 679, 36-43.
  • Chen, X., Y. Zhu, T. Zhu, P. Song, J. Guo, Y. Zhong, L. Gui, and M. Li, Vasa identifies germ cells in embryos and gonads of Oryzias celebensis. Gene, 2022. 823: p. 146369.
  •  Sun, B., L. Gui, R. Liu, Y. Hong, and M. Li, Medaka oct4 is essential for gastrulation, central nervous system development and angiogenesis. Gene, 2020. 733: p. 144270”.

Round 2

Reviewer 1 Report

The authors have sufficiently addresses my comments and concerns. However, the answer of comment no. 45 might be still incorrect. It should be "(A-C) View under the low magnification" rather than ". (A-F) View under the low magnification". Please check it. 

Author Response

    Many thanks for your careful reading , we have changed from (A-F) to (A-C).

We are very grateful  for your kind help!